# Glycerol-Free Equilibration with the Addition of Glycerol Shortly before the Freezing Procedure: A Perspective Strategy for Cryopreservation of Wallachian Ram Sperm

**DOI:** 10.3390/ani13071200

**Published:** 2023-03-29

**Authors:** Anežka Málková, Filipp Georgijevič Savvulidi, Martin Ptáček, Karolína Machová, Martina Janošíková, Szabolcs Nagy, Luděk Stádník

**Affiliations:** 1Department of Animal Science, Faculty of Agrobiology, Food and Natural Resources, Czech University of Life Sciences Prague, Kamýcká 129, 165 00 Prague, Czech Republic; malkovaa@af.czu.cz (A.M.);; 2Department of Genetics and Breeding, Faculty of Agrobiology, Food and Natural Resources, Czech University of Life Sciences Prague, Kamýcká 129, 165 00 Prague, Czech Republic; 3Institute of Animal Sciences, Hungarian University of Agriculture and Life Sciences, Georgikon Campus, H8360 Keszthely, Hungary

**Keywords:** cryopreservation capacity, sperm motility, plasma membrane and acrosome intactness, mitochondrial membrane potential

## Abstract

**Simple Summary:**

This study investigated the effect of cryoprotectant glycerol addition in different phases of sperm equilibration on sperm motility and viability parameters of thawed ram spermatozoa. The purpose of this study was to compare two strategies of glycerol addition for sperm cryo-preservation. The first strategy included the use of a glycerol-free extender for the procedure of glycerol-free equilibration and chilling, with the glycerolation of the extender by 6% glycerol shortly before sperm slow freezing. The second strategy included the use of a freezing extender already glycerolated by 6% glycerol before the equilibration and chilling of sperm and following slow freezing. The obtained results were statistically significant and demonstrated that the glycerol-free equilibration with the addition of glycerol at least 10 min before freezing is the best strategy for cryopreservation of Wallachian ram sperm. This represents a perspective strategy in the cryopreservation process for local sheep breeds and indicates a potential strategy to test for ram cryopreservation in a generalized manner.

**Abstract:**

This study investigated the effect of glycerol added in different phases of sperm equilibration on CASA and flow cytometry parameters of thawed ram spermatozoa. Sperm was collected from adult Wallachian rams. The freezing extender was glycerol-free ANDROMED^®^ (Minitub GmbH, Tiefenbach, Germany) supplied by 6% exogenous glycerol at different stages of the cryopreservation process. The purpose of this study was to compare two strategies of glycerol addition for sperm cryopreservation. The first strategy included the use of a glycerol-free extender for the procedure of glycerol-free equilibration and chilling, with the glycerolation of the extender by 6% glycerol shortly before sperm slow freezing (GFA). The second strategy included the use of a freezing extender already glycerolated by 6% glycerol before the equilibration and chilling of sperm and following slow freezing (GA). Sperm samples were analyzed after equilibration (but before freezing) and after thawing (at T0, T1 h, and T2 h time points). iSperm^®^ mCASA (Aidmics Biotechnology Co., LTD., Taipei, Taiwan) was used for the evaluation of sperm kinematics. Flow cytometry was used to measure sperm viability (plasma membrane/acrosome intactness) and mitochondrial membrane potential. The obtained results significantly demonstrated that the glycerol-free equilibration with the addition of glycerol shortly before freezing is a perspective strategy for cryopreservation of Wallachian ram sperm.

## 1. Introduction

The Wallachian sheep is an original breed of the Czech Republic with a triple-yield purpose (milk, meat, and wool). The breed is resistant to harsh climatic conditions, historically managed under the Carpathian grazing system. Wallachian sheep are classified among genetic resources and protected by the Czech national rescued program of threatened species for their historical and genetic uniqueness. Machova et al. [1] estimated the average inbreeding coefficient at 5% and the average relatedness coefficient at 9% in a population of Wallachian sheep. The inbreeding coefficient has a cumulative effect and could reach as high an inbreeding coefficient as 7% over the next 30 years, as Machova et al. [1] have added. Sonesson et al. [2] hypothesized that inbreeding can be reduced by a correctly designed strategy of frozen semen insemination when different generations are introduced in a closed small-scaled population. The cryopreservation process of the ram ejaculate represents a necessary part of the rescue program for eliminating the negative effect of inbreeding and for establishing an adequate reservoir of genetically valuable sires. Unfortunately, there is a very low tolerance of ram spermatozoa to cryopreservation; therefore, optimizing this procedure is necessary for the rescue program of this unique breed [3,4].

Cryoprotective substances, used for the cryopreservation of sperm, prevent damage to cell structures such as DNA, acrosomes, axonemes, mitochondria, or plasma membranes [5]. Cryoprotective agents (CPAs), extracellular or intracellular, non-penetrating or infiltrating the cell’s intracellular space [6], are a highly desirable ingredient in the freezing of semen and have a practical impact in small ruminants’ semen cryopreservation [7]. Glycerol represents the most common CPA, used as a protective component of the diluent [5,7]. The level of glycerol concentration in the extender has limitations for individual animal species [7], as glycerol naturally causes a reduction in DNA integrity [8], osmotic damage [9,10,11], or cell structural damage [12]. This toxicity of glycerol was observed in a higher concentration than 8% [13,14,15]. For that reason, a 3% to 8% range of glycerol concentration is recommended [16,17,18]. The degree of glycerol toxicity can, however, be potentially eliminated by reduced exposure of glycerol to sperm cells during the process of equilibration. This assumption follows previous studies published by Salamon [19]. The addition of glycerol after equilibration just before starting the freezing procedure should be adequate for ram sperm cryopreservation, as its penetration ability into the sperm cell takes only several seconds [20,21,22].

This study aimed to investigate different methods of glycerol addition during a freezing procedure on the kinematic characteristics and integrity of sperm organelles of equilibrated and frozen–thawed semen. A wider aim of this study was to suggest an improved strategy for the cryopreservation process of Wallachian ram spermatozoa.

## 2. Materials and Methods

### 2.1. Reagents and Solutions

The solution of Dulbecco’s phosphate-buffered saline (PBS) (–Ca/–Mg) was purchased from Sigma-Aldrich (Merck KGaA, Darmstadt, Germany). The freezing extender was glycerol-free ANDROMED^®^ (Minitüb GmbH, Tiefenbach, Germany). The content of glycerol supplementation in the extender was 6% (this concentration of glycerol is used in commercially available ANDROMED^®^ extender, as reported by Akçay et al. [23]). The glycerol reagent G5516 for molecular biology was obtained from Sigma-Aldrich (Merck KGaA, Darmstadt, Germany). Hoechst-33342 (H-342) and propidium iodide (PI) for flow cytometry were purchased from Sigma Aldrich (St. Louis, MO, USA), and lectin PNA from Arachis hypogaea (PNA-FITC) and MitoTracker™ Deep Red (MTR DR) were obtained from Thermo Fisher Scientific (Waltham, MA, USA).

### 2.2. Ethical Statements

All methods were carried out following the relevant guidelines and regulations. All experimental protocols were approved by the review board of the expert commission ensuring the welfare of experimental animals at the Czech University of Life Sciences in Prague. Except for semen collection using an artificial vagina, the present study did not involve any in vivo animal experiments; further, no pharmacological procedures, surgical procedures, animal pathogen infection, or euthanasia were used in the present study. Therefore, the review board of the expert commission ensuring the welfare of experimental animals at the Czech University of Life Sciences in Prague considers that this type of research does not fall under the legislation for the protection of animals used for scientific purposes (Act no. 246/1992 Sb.). It considers that this type of research has no impact on animal welfare because only non-experimental agricultural practices (following Act no. 166/1999 Sb. and Act no. 154/2000 Sb.) were used during the study.

### 2.3. Animal Management

Owing to difficulties with the availability of sires of the endangered Wallachian local breed, a limited number (two) of mature purebred Beskyd bloodline and Juras bloodline Wallachian rams were used in the present study. Rams with typical exterior signs of the breed and with an excellent breeding history were transported from the area of the Beskids Mountains (CZ). Afterwards, animals were kept at the Demonstration and Experimental Centre of the Czech University of Life Sciences in Prague (50°07′47.6″ N 14°22′07.0″ E), at the nearest distance to the laboratory of flow cytometry. This was done with a special intention to reduce any variability due to the possible delay between the collection of semen and its subsequent evaluation. Moreover, this enabled the use of rams in the same health condition under strict veterinary inspection, in the same breeding conditions, under the same feeding ration, and with the semen collected and processed in the same manner and over the same time duration. It is important to also mention that the Demonstration and Experimental Centre of the Czech University of Life Sciences in Prague provided us with the possibility to only keep a limited number of animals. As we were aware of the possibility of using only a limited number of animals, the repetition of semen collection from both rams was highly increased in the present study (as described in Section 2.4 “Semen collection“). Both rams had free-range access to barn shelters and free-range access to a fenced grazing area. The feed ration consisted of pasture and hay from its own production (ad libitum), as well as commercially available concentrated pellets (1580 OVCE UNI, De Heus a.s., Bučovice, Czech Republic; BMK TEX O, VVS Verměřovice s.r.o., Czech Republic), regulated according to the animal body condition score at an amount of approximately 0.25–1 kg/pcs/day. Mineral licks (Solsel, Mikrop Čebín a.s., Czech Republic) and water were available ad libitum. Animal health and nutritional status (BCS 3.5–4) were continuously monitored [24].

### 2.4. Semen Collection

Semen was naturally collected on a dummy (heat-identified ewe) using the artificial vagina (Minitub GmbH, Tiefenbach, Germany), prepared according to the manufacturer’s instructions. Ejaculate samples were taken at regular intervals on 17 individual sampling days for flow cytometry and on 14 days for the evaluation of the CASA method. Mainly, the first collected ejaculate was taken and processed from each ram per sampling day. According to our observations, a similar sperm quality across ranks of semen collection was detected for Wallachian rams (data not published). Shortly after collection, each ejaculate was divided into two balanced parts and each part was extended to 1:4 (*v*/*v*) with the semen extender ANDROMED^®^ (Minitub GmbH, Tiefenbach, Germany) glycerolated according to two strategies chosen for extender glycerolation: GFA (the use of a glycerol-free extender for the procedure of glycerol-free equilibration and chilling, with the glycerolation of the extender by 6% glycerol shortly before sperm slow freezing), or GA (the use of a freezing extender already glycerolated by 6% glycerol before the equilibration and chilling of sperm and following slow freezing). Extended sperm samples were transported to the laboratory in a thermobox at 25 °C for further processing.

### 2.5. Semen Processing and Equilibration

At the laboratory, the sperm concentration of each extended sperm sample was assayed using a precalibrated Genesys™ 10vis spectrophotometer (Thermo Fisher Scientific, Waltham, MA, USA). To eliminate any ram-to-ram variation, sperm was collected from both animals (separately for the GFA and GA glycerolation strategy) and were pooled 1:1 (*v*/*v*), obtaining a final concentration of 200 × 10^6^ sperm cells/mL [17]. Sperm was filled into 0.25 mL French straws (IMV Technologies, L’Aigle, France) at a temperature of 25 °C. The straws were sealed with sealing powder (IMV Technologies, L’Aigle, France) and equilibrated for two hours [25] in a cold room at 6 °C.

### 2.6. Sperm Freezing and Thawing

After equilibration, GFA sperm was supplemented in one step with pre-cooled glycerol (6%) shortly before freezing. GFA sperm was frozen within 10 min after glycerol supplementation using the freezing equipment described by Ptáček et al. [26]. A freezing protocol (a curve) was used, optimized previously for freezing sperm from Wallachian ram [3]. GA sperm was frozen using a similar freezing method as for GFA sperm. After freezing, the straws were stored in liquid nitrogen for at least 24 h. Straws were thawed by plunging in a water bath (thawing temperature and duration: 38 °C and 30 s, respectively). No thawing diluent was used.

### 2.7. Sperm Motility Evaluation by iSperm^®^ mCASA

Sperm was evaluated after equilibration (EQ) and shortly after thawing (T0). To determine motility and kinematic parameters, the mobile computer-assisted sperm analyzer iSperm^®^ mCASA (Aidmics Biotechnology, Taipei, Taiwan) was used, following the manufacturer’s instructions for software use. At least three fields of each iSperm sample chip were captured at 22 frames/s at 30 fps. The iSperm setting was as follows for kinetic parameters: VAP (average path velocity, μm/s) ≥ 50 μm/s and STR (straightness, %) ≥ 70% considered progressive (%); VAP (μm/s) ≥ 20 μm/s and VSL (straight-line velocity, μm/s) ≥ 0 μm/s considered motile (%). Other measured properties were curvilinear velocity (VCL, μm/s) and linearity (LIN, %).

### 2.8. Flow Cytometry Assessment

Sperm samples were evaluated by flow cytometry after EQ, shortly after thawing (T0), one hour after thawing (T1), and two hours after thawing (T2). Sperm at T1 and T2 were kept at 38 °C before flow cytometry analysis. Before the evaluation in a 96-well plate, samples were diluted in PBS with dyes to a final concentration of 20 × 10^6^ spermatozoa/mL and then incubated for 10 min at 38 °C in the dark. Fluorescent dyes were prepared on the same day of evaluation. The samples were stained with fluorescent dyes as follows (final concentrations are given): 10 μg/mL H-342 for the identification of DNA content; 8 μg/mL PI to determine plasma membrane damage; 0.5 μg/mL PNA-FITC to assess acrosome damage; and 80 nM MTR DR to assess mitochondrial activity (mitochondrial membrane potential—MMP) [27]. For evaluation, a digital flow cytometer NovoCyte 3000 (Acea Biosciences, Agilent, Santa Clara, CA, USA) was used. The flow cytometer was equipped with a set of optimal bandpass filters as well as set of solid-state lasers: a violet laser (405 nm, 50 mW) for H-342 excitation, a blue laser (488 nm, 60 mW) for PI and PNA-FITC excitation, and a red laser (640 nm, 40 mW) for MTR-DR excitation. H-342 can be successfully excited with a violet (405 nm) laser [28]. The calibration of the instrument was performed using calibration beads (NovoCyte, QC Particles, Agilent Technologies, Santa Clara, CA, USA) before the samples were used. The samples were analyzed at a low speed and at least 10,000 events identified as sperm cells were evaluated from each sample. For automated cytometer setup, performance tracking and data acquisition were used with NovoExpress software, v1.3.0 (Acea Biosciences, Agilent, Santa Clara, CA, USA). The data were saved and subsequently analyzed by the same software (Figure 1). No compensation was required with the optical filter settings used [3].

### 2.9. Statistical Analysis

Statistical analyses were performed using SAS/ STAT^®^ v9.4. (SAS Institute Inc., Cary, NC, USA) and the generalized linear model (GLM) procedure. The following statistical models were designed for the estimation of CASA parameters:Y_ijk_ = µ + DAY_i_ + VAR_j_ + e_ijk_ (for equilibrated sperm cells only);Y_ijk_ = µ + DAY_i_ + VAR_j_ + b_1−7_ × EQ+ e_ijk_ (for cryopreserved sperm cells only).

Here, Y_ijk_ is an evaluated trait (MOT, PROG, VAP, STR, VSL, LIN, and VCL); µ is the mean of the evaluated trait; DAY_i_ is a fixed effect of the day of semen collection (i = 4, n = 8; i = 5, n = 8; i = 6, n = 16; i = 7, n = 16; i = 8, n = 8; i = 9, n = 8; i = 10, n = 16; i = 11, n = 8; i = 12, n = 8; i = 13, n = 8; i = 14, n = 8; i = 15, n = 8; i = 16, n = 8; and i = 17, n = 9); VAR_j_ is a fixed effect of glycerol addition variation (j = GA, n = 76; j = GFA, j = 61); b_1-7_ × EQ is a particular kinematic parameter of equilibrated spermatozoa as a covariate for the correction of the same kinematic parameter after thawing; and e_ijk_ is the residual error.

The following statistical models were designed for the estimation of flow cytometry parameters:Y_ijk_ = µ + DAY_i_ + VAR_j_ + DAY × VAR_ij_ + e_ijk_ (for equilibrated sperm cells only);Y_ijk_ = µ + DAY_i_ + VAR_j_ + DAY × VAR_ij_ + b_1−2_×EQ + e_ijk_ (for cryopreserved sperm cells only).

Here, Y_ijk_ is an evaluated trait (total cell viability and mitochondrial membrane potential measured in equilibrated cells, immediately after thawing or after 1 h or after 2 h of incubation); µ is the mean of the evaluated trait; DAY_i_ is a fixed effect of the day of semen collection (i = 1, n = 16; i = 2, j = 8; i = 3, n = 16; i = 4, n = 12; i = 5, n = 12; i = 6, n = 16; i = 7, n = 6; i = 8, n = 8; i = 9, n = 6; i = 10, n = 8; i = 11, n = 6; i = 12, n = 6; i = 13, n = 6; i = 14, n = 6; i = 15, n = 6; i = 16, n = 6; and i = 17, n = 6); VAR_j_ is a fixed effect of glycerol addition variation (j = GA, n = 75; j = GFA, j = 75); DAY × VAR_ij_ is an interaction effect of the day of semen collection and glycerol addition variation; b_1−2_ × EQ is a particular flow cytometer parameter of equilibrated spermatozoa as a covariate for the correction of the same flow cytometer parameter after thawing, as well as after 1 h and after 2 h of incubation; and e_ijk_ is the residual error.

Significant differences were estimated by Tukey–Kramer adjustment at a significance level of 0.05.

## 3. Results

### 3.1. CASA Parameters

All statistical models for equilibrated and frozen–thawed sperm kinematic CASA parameters were significant. Supporting data, covering the significance of particular factors in model equations, are reported in Table 1.

Figure 2 demonstrates the results of the GFA and GA methods used for equilibrated sperm. All monitored parameters differed significantly, except for VAP_EQ_. Surprisingly, significantly higher values were detected for the GA method over the GFA variation for equilibrated sperm before freezing. This involved MOT_EQ_ (approximately 1/3 decrease in GFA over GA) and PROG_EQ_ (more than 2/3 decrease in GFA over GA) in particular.

Importantly, the results of GFA were not significantly worse than those of GA immediately after thawing, as reported in Figure 3. Moreover, the beneficial effect of spermatozoa freezing using the GFA variation was supported by 3.25% (*p* < 0.05) higher MOT and 8.7 μm/s (*p* < 0.05) higher VAP.

### 3.2. Flow Cytometry Parameters

Models were significant for all of the evaluated flow cytometric parameters at all of the evaluated time intervals. The results of GA and GFA analyses, supported by the significance of particular factors in model equations, are reported in Table 2.

The GA method was detected to have 23.49% higher viability of equilibrated sperm over the GFA method. A significantly higher mitochondrial membrane potential (+9.26%, *p* < 0.05) was noticed for spermatozoa equilibrated in GA as well.

In general, both flow cytometric parameters in GFA surpassed those in GA after thawing and a higher stability was maintained in GFA throughout sperm incubation. This can be summarized and identified in Table 2. Sperm viability immediately after thawing was 4.74% higher for GFA, and it was even 10.92% higher after 2 h of incubation. It is considerable that a greater than 12.00% decrease in viability was observed for GA throughout the incubation period (from 22.38% at PAI_T0_ to 9.61% at PAI_T2_). The decrease for GFA reached only 6.59% (from 27.12% to 20.53%) during this time. The mitochondrial membrane potential was significantly lower for GA immediately after thawing and, furthermore, evidenced a decrease of greater than 2.5 times during the first hour of incubation, as well as a decrease of greater than 1.5 times during the second hour. As a result, the mitochondrial membrane potential was 11.81% (*p* < 0.05) and 4.23% (*p* < 0.05) higher for GFA than for GA during first and second hours of sperm incubation, respectively.

## 4. Discussion

More than 50 years ago, the first studies investigated whether the presence of glycerol in the non-frozen state may have a toxic effect on livestock sperm [29]. Glycerol can represent a stress factor for spermatic cells, thus decreasing the conception rate after thawing. Sharafi et al. [30] have assessed that glycerol impacts plasma membrane coats, such as glycocalyx, glycoproteins, and glycolipids, and may lead to membrane deterioration and increase the cytosol viscosity. Additionally, glycerol may change the polymerization and depolymerization of α and β tubulins, the major proteins of the microtubules in sperm tails. Slavík [31] stated that fresh sperm incubated for 30–90 min with glycerol led to an increase in the sperm penetration of zona-free hamster eggs in a short period. This was caused by a glycerol-induced acrosome reaction, which could be the reason for the low conception rate after cervical insemination as well. Previously published studies have demonstrated that, after equilibration, incubating sperm with glycerol at 5 °C for more than several hours leads to evident sperm damage [32,33]. Therefore, a perspective approach of how to diminish the toxic effect of glycerol, but still maintain its cryoprotective effect, was developed by other authors—namely, the addition of glycerol to sperm shortly before freezing, after equilibration. Jones [34] states that the addition of glycerol for 10 min before ram sperm freezing is sufficient. A range of authors have observed that the risk of cell damage due to the penetration of glycerol into spermatozoa was reduced with a short period of glycerol exposure to sperm cells [20,29,35]. Jones [34] concluded that the addition of glycerol for 10 min before ram sperm freezing is sufficient.

In the present study, however, we did not observe reduced sperm motility after two hours of equilibration of sperm with glycerol when using the GA strategy. On the contrary, we found significantly higher motility values of MOT_EQ_ and PROG_EQ_ for the GA method over the GFA method for equilibrated sperm before freezing. Furthermore, we were not able to demonstrate any significant toxic effect of glycerol in the non-frozen state upon ram spermatozoa by means of flow cytometry (on the contrary, we found a higher viability and higher mitochondrial membrane potential of equilibrated sperm in the GA method over the GFA method). Our results are in line with a previously published study reporting that glycerol can be used by a sperm cell as a source of energy after transporting through sperm aquaporins [36]. We hypothesized that ram sperm cells equilibrated in the presence of glycerol might better preserve their cellular fitness, as opposed to sperm cells equilibrated without glycerol. Interestingly, several authors reported previously that, when using fresh or chilled semen, it is advisable to use glycerol as a protective agent to maintain cell motility in the ejaculate and to improve fertilization ability in artificial insemination [37,38].

The process of cryopreservation causes damage to sperm cells as a result of intracellular ice crystallization and other factors. This cryodamaging phenomenon is well described [39]. After thawing, the seminal parameters at time T0, T1, and T2 followed a very normal trend (higher at T0 and lower at T2). Therefore, in the present study, we focused only on monitoring the qualitative differences between the GFA and GA methods in individual phases (T0, T1, and T2) of the cryopreservation process. A possible reason for the observed decrease in the viability of the GA method of glycerol addition after thawing could still be due to the longer duration of the action of glycerol during the two-hour equilibration in cells before freezing. We hypothesized that the accumulation of other degradable glycerol compounds in the sperm, and thus the increasing toxicity of glycerol after thawing, might be the reason [40,41].

In terms of our results after thawing (and in spite of the results obtained in the present study after equilibration, but before freezing), the glycerolation just before freezing significantly improved the viability, the integrity of the sperm plasma membrane, and the mitochondrial membrane potential.

## 5. Conclusions

The addition of glycerol at least 10 min before freezing Wallachian ram sperm improved the CASA and flow cytometry parameters after sperm thawing. This represents a perspective strategy in the cryopreservation process for local sheep breeds and indicates a potential strategy to test for ram cryopreservation in a generalized manner.

Future studies are needed with a focus on better understanding the effect of glycerol on ram sperm cells. Particularly, research suggesting advanced strategies for minimizing the cytotoxicity of glycerol, but with full retention of its cryoprotective abilities with respect to our results, should follow. Considering the limitations of our study, the obtained results should be generalized to a larger population of rams, as well as to a different breed, and/or should be subsequently verified by artificial insemination.

## Figures and Tables

**Figure 1 animals-13-01200-f001:**
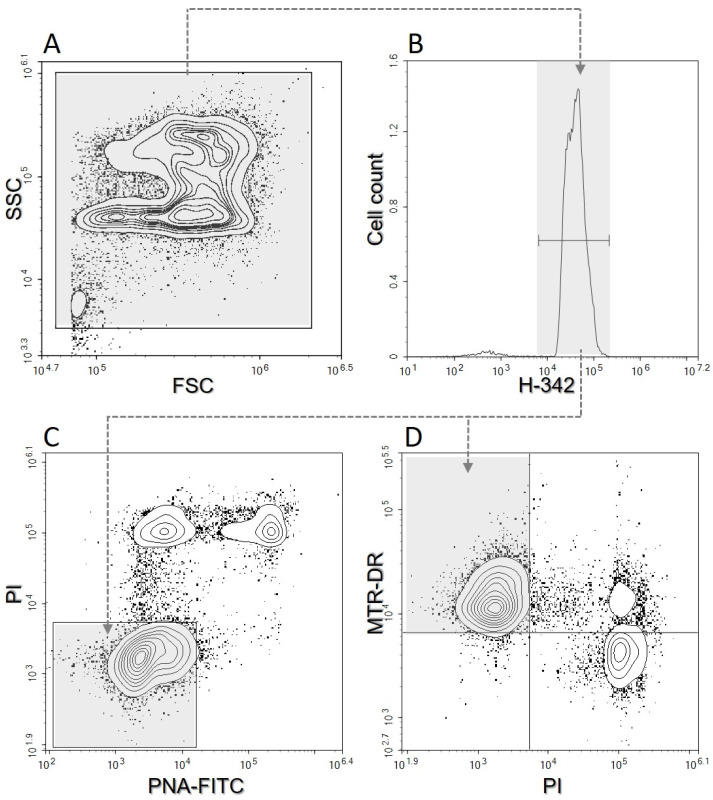
Illustration of the flow cytometric gating strategy. (**A**)—Side scattering vs. forward scattering. (**B**)—Cell count vs. Hoechst-33342 (H-342). (**C**)—Propidium Iodide (PI) vs. lectin PNA (PNA-FITC); the PAI (plasma membrane and acrosome intact sperm) region is shown in grey. (**D**)— MitoTracker™ Deep Red (MTR-DR) vs. PI; the MMP (mitochondrial membrane potential) region is shown in grey.

**Figure 2 animals-13-01200-f002:**
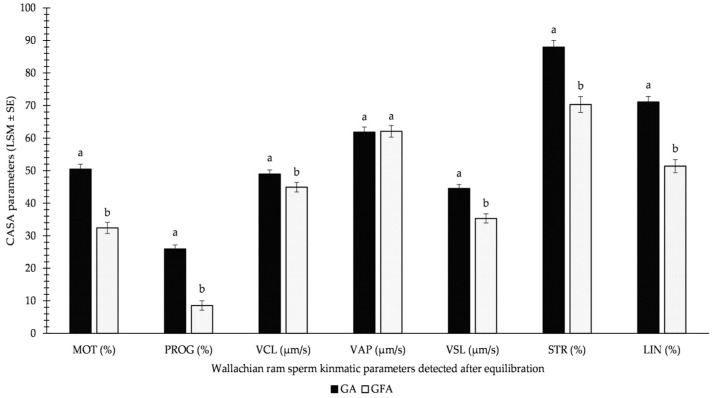
Influence of GA (glycerolation during the whole equilibration process) vs. GFA (glycerolation 10 min before the freezing procedure) on CASA parameters in Wallachian ram sperm detected immediately after equilibration; MOT: total motility; PROG: progressive motility; VCL: curvilinear velocity; VAP: average path velocity; VSL: straight-line velocity; STR: straightness; LIN: linearity; a–b: different letters within particular parameters indicate significant differences at a *p* < 0.05 level of significance.

**Figure 3 animals-13-01200-f003:**
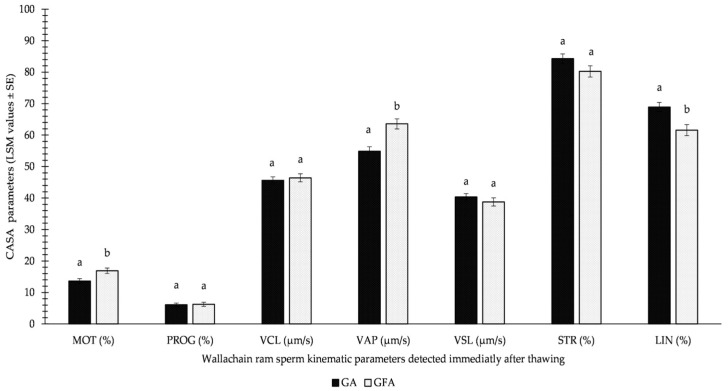
Influence of GA (glycerolation during the whole equilibration process) vs. GFA (glycerolation 10 min before the freezing procedure) on CASA parameters in Wallachian ram sperm detected immediately after thawing; MOT: total motility; PROG: progressive motility; VCL: curvilinear velocity; VAP: average path velocity; VSL: straight-line velocity; STR: straightness; LIN: linearity; a–b: different letters within particular parameters indicate significant differences at a *p* < 0.05 level of significance.

**Table 1 animals-13-01200-t001:** Significance of factors used in statistical models for CASA parameters.

	b_1–7_ × EQ	DAY	VAR
MOT_EQ_	---	***	***
PROG_EQ_	---	***	***
VCL_EQ_	---	***	*
VAP_EQ_	---	***	n.s.
VSL_EQ_	---	***	***
STR_EQ_	---	***	***
LIN_EQ_	---	***	***
MOT_T0_	**	***	**
PROG_T0_	*	***	n.s.
VCL_T0_	***	***	n.s.
VAP_T0_	***	***	***
VSL_T0_	***	***	n.s.
STR_T0_	***	*	n.s.
LIN_T0_	***	**	***

b_1–7_ × EQ: particular kinematic parameter of equilibrated spermatozoa as a covariate for the correction of the same kinematic parameter after thawing; DAY: effect of fixed day semen collection; VAR: fixed effect of method of glycerol addition; MOT_EQ,T0_: motility after equilibration (EQ) and after thawing (T0); PROG_EQ,T0_: progressive motility (EQ) after equilibration and after thawing (T0); VCL_EQ,T0_: curvilinear velocity after equilibration (EQ) and after thawing (T0); VAP_EQ,T0_: average path velocity after equilibration (EQ) and after thawing (T0); VSL_EQ,T0_: straight-line velocity after equilibration (EQ) and after thawing (T0); STR_EQ,T0_: straightness after equilibration (EQ) and after thawing (T0); LIN_EQ,T0_: linearity after equilibration (EQ) and after thawing (T0); *—*p* < 0.05; **—*p* < 0.01; ***—*p* < 0.001; n.s.—non-significant.

**Table 2 animals-13-01200-t002:** Significance of factors used in statistical models for flow cytometric parameters.

	GA (%)	GFA (%)	Significance of Factors in Statistical Model			
			b_1–2_ × EQ	DAY	VAR	DAY × VAR
PAI_EQ_	60.48 ± 1.61 ^a^	36.99 ± 1.61 ^b^	---	***	***	***
PAI_T0_	22.38 ± 0.87 ^a^	27.12 ± 0.82 ^b^	***	***	***	***
PAI_T1_	10.92 ± 0.74 ^a^	22.55 ± 0.69 ^b^	***	***	***	***
PAI_T2_	9.61 ± 0.68 ^a^	20.53 ± 0.63 ^b^	***	***	***	***
MMP_EQ_	64.20 ± 1.75 ^a^	54.92 ± 1.75 ^b^	---	***	***	***
MMP_T0_	36.67 ± 1.40 ^a^	41.03 ± 1.30 ^b^	***	***	*	***
MMP_T1_	7.16 ± 0.82 ^a^	18.97 ± 0.77 ^b^	*	***	***	***
MMP_T2_	7.49 ± 0.75 ^a^	11.72 ± 0.71 ^b^	***	***	***	***

GA: glycerolation during the whole equilibration process; GFA: glycerolation 10 min before the freezing procedure; b_1-2_×EQ: particular flow cytometer parameter of equilibrated spermatozoa as a covariate for the correction of the same flow cytometer parameter after thawing (T0), as well as after 1 h (T1) and after 2 h (T2) of incubation; DAY: effect of fixed day semen collection; VAR: fixed effect of the method of glycerol addition (GA vs. GFA); DAY × VAR: interaction of the effect of the day of semen collection and method of glycerol addition; PAI_EQ,T0,T1,T2_: viable sperm (cells with plasma membrane and acrosome intactness) after equilibration (EQ), after thawing (T0), after 1 h of incubation (T1), and after 2 h of incubation (T2); MMP_EQ,T0,T1,T2_: high mitochondrial membrane potential after equilibration (EQ), after thawing (T0), after 1 h of incubation (T1), and after 2 h of incubation (T2); ---: non estimated parameter; *—*p* < 0.05; ***—*p* < 0.001; n.s.—non-significant; ^a–b^ different letters indicate differences between the methods of glycerol addition.

## Data Availability

All data are presented in the manuscript.

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
