# Peer review of "Glycerol-Free Equilibration with the Addition of Glycerol Shortly before the Freezing Procedure: A Perspective Strategy for Cryopreservation of Wallachian Ram Sperm"

_animals, 2023, doi:10.3390/ani13071200_

Round 1

Reviewer 1 Report

The objective of this work is the optimization of the protocols for freezing ram semen of a specific breed. They focus on studying the effect of glycerol on the cryoresistance of sperm.

The argument is adequate due to the serious situation of this breed as indicated by the authors. But the fact that they only get two males to carry out the study leaves it very limited.

Removing the lack of number of males, the experimental design is adequate, although sperm analysis could have been increased and mainly included artificial insemination.

The results are consistent with the experimental development. Likewise, the results are well presented.

The discussion would have to be modified. The first paragraphs seem like a second introduction... commenting on the situation and problems of the freezing.... why the work is proposed.....justifies the objectives again... The authors should synthesize the results, why do you get these results, what are the causes... you can argue this based on what you comment in these paragraphs, but giving them a more discussion orientation, not exposition. Line 325-328 expand this argument. Nothing is explained about the seminal parameters at time T1 and T2

Other comment:

-        L138-140.  Does not placing the straws from 25ºC to 6ºC produce a sudden drop in temperature? Is this really the protocol?

-        L136. “….To eliminate ram-to-ram variation, extended sperm collected from both animals were diluted with relevant extender,to obtain a final concentration of 200 × 106 sperm cells/ml [17] and further pooled 1:1 (vol/vol)” …. I don't understand what you mean with this phrase.

-        L308 and 318…” The above-mentioned results…” ; “Due to the abovementioned contradictions ...” … choose.

Reviewer 2 Report

In general, it is a very interesting study, since it is necessary to investigate new freezing protocols that allow improving the seminal quality after thawing and, of course, fertility after AI, especially if it concerns breeds in endangered of extinction. However, it would be necessary to make some comments and clarify some issues that I proceed to detail below:

L65. You mean "semen in bulls" or "semen in ram"?The article you refer to is about semen in ram.

L108. The number of males used for the study is very low. It would be necessary in the future to carry out this experiment with other males to reinforce the results obtained.

L119. For the semen collection of the males, was a dummy or an estrogenized female used? How many ejaculates were collected per ram and day? Is extender used for semen collection?

L141. What is the reference of the freezing protocol that it was used? When glycerol is added, is it added in several parts separated by a few minutes to gradually increase the osmotic pressure to dehydrate the cells and thus avoid damage to cell membranes? For thawing, is a thawing diluent used to dilute the glycerol and its toxic effects, while maintaining the appropriate conditions for sperm survival?

Reviewer 3 Report

The work presented is interesting and well developed and explained. It is always necessary to carry out studies that allow the conservation of breeds in danger of extiction. On the other hand, after reading the work, I would like to indicate that tables 1 and 2 should be reviewed by default. Otherwise, the work is correct.

Reviewer 4 Report

In your manuscript "Glycerol-free equilibration with the addition of glycerol shortly before freezing procedure: a perspective strategy for cryopreservation of Wallachian ram sperm" you present a study in which you compare two different ejaculate-processing approaches , one with glycerol present in the buffers right from the start, and one in which glyerol is added later, just before freezing, on ejaculates from two Wallachian ram (of two bloodlines, one of the Beskyd bloodline and one of the Juras bloodline). You observe no toxic effect of glycerol if present right from the start, and do observe some slightly improved sperm parameters after thawing when glycerol is added later.

The mansucript is more a technical note than a full scientific manuscript. With only two different procedures assessed on semen of only two rams, and with each being from a distinct blood line, the oberservations made here, albeit interesting, can only be regarded as a short tecnical comment. The study design, as stated above is flawed, although all technical aspects of the semen analysis are in order, and the presentation of the data is clear and concise. Also, I believe that this manuscript is of interest only to a small fraction of the readers of "Animals". 

Round 2

Reviewer 4 Report

The manuscript has improved due to the changes in e.g. the discussion. However, I still stick to my previous comments, that this work is based on too few animals and therefore lacks the merrits of a full scientific paper.